# An Overview of Antibody Conjugated Polymeric Nanoparticles for Breast Cancer Therapy

**DOI:** 10.3390/pharmaceutics12090802

**Published:** 2020-08-25

**Authors:** Alberto Juan, Francisco J. Cimas, Iván Bravo, Atanasio Pandiella, Alberto Ocaña, Carlos Alonso-Moreno

**Affiliations:** 1Oncología traslacional, Unidad de Investigación del Complejo Hospitalario Universitario de Albacete, 02008 Albacete, Spain; alberto.juan@uclm.es (A.J.); franciscojose.cimas@uclm.es (F.J.C.); 2Centro Regional de Investigaciones Biomédicas, Unidad NanoCRIB, 02008 Albacete, Spain; ivan.bravo@uclm.es; 3School of Pharmacy, University of Castilla-La Mancha, 02008 Albacete, Spain; 4Centro de Investigación del Cáncer-CSIC, IBSAL- Salamanca and CIBERONC, 37007 Salamanca, Spain; atanasio@usal.es; 5Experimental Therapeutics Unit, Hospital clínico San Carlos, IdISSC and CIBERONC, 28040 Madrid, Spain

**Keywords:** breast cancer, antibody conjugated, polymeric nanoparticles, antibody drug conjugates, antibody drug conjugate nanoparticles

## Abstract

Nanoparticles (NPs) are promising drug delivery systems (DDS) for identifying and treating cancer. Active targeting NPs can be generated by conjugation with ligands that bind overexpressed or mutant cell surface receptors on target cells that are poorly or not even expressed on normal cells. Receptor-mediated endocytosis of the NPs occurs and the drug is released inside the cell or in the surrounding tissue due to the bystander effect. Antibodies are the most frequently used ligands to actively target tumor cells. In this context, antibody-based therapies have been extensively used in HER2+ breast cancer. However, some patients inherently display resistance and in advanced stages, almost all eventually progress. Functionalized NPs through conjugation with antibodies appear to be a promising strategy to optimize targeted therapies due to properties related to biocompatibility, suitable delivery control and efficiency of functionalization. This review is focused on the different strategies to conjugate antibodies into polymeric NPs. Recent antibody conjugation approaches applied to the improvement of breast cancer therapy are highlighted in this review.

## 1. Introduction

Being a woman is the main risk factor for developing breast cancer and, unfortunately, in advanced stages, the disease remains incurable. According to immunohistochemical and transcriptomic criteria [1], breast cancer can be divided into three subtypes: (1) tumors that express hormonal receptors, so-called luminal; (2) cancers that overexpress the transmembrane tyrosine kinase HER2; and (3) tumors that do not overexpress any of the above-mentioned proteins, the so called triple negative breast cancers (TNBC). Among the different subtypes, those that overexpress HER2 (HER2+) occur in approximately one out of five women diagnosed with breast cancer.

Although HER2 expression has historically been associated with poor outcome, over the last 20 years, different therapies have been approved, improving the prognosis of patients with this type of tumor. Antibodies against the extracellular domain of the receptor (trastuzumab and biosimilars, pertuzumab or T-DM1) and small molecule inhibitors of the kinase activity of the receptor (lapatinib, tucatinib or neratinib) have reached the clinic. Despite the impact of trastuzumab-based therapies, there are patients inherently resistant to the drug. Moreover, in advanced disease states, almost all patients will progress on trastuzumab. The TNBC subtype is characterized by the absence of HER2 overexpression and undetectable levels of estrogen and progesterone receptors. Even though it only represents 15% of breast tumors, the prognosis is poor due to the limited therapeutic options that are currently available. In this context, the development of novel and improved therapeutics is a primary objective and finding targeted therapies is a very promising approach. In fact, antibody drug conjugates (ADCs) are very successful targeted delivery systems (eight ADCs were approved by the United States Food and Drug Administration (FDA) in recent years) [2,3,4,5]. Accordingly, the use of nanomedicines for detecting and targeting transmembrane receptors in cancer cells can also be an attractive approach for the treatment of cancer, as it acts on cancer cells in a specific manner, avoiding undesirable effects to normal tissue. The conjugation of nanocarriers, such as nanoparticles (NPs), with antibodies to generate targeted therapies has been proposed as a novel strategy for the treatment of cancer [6,7,8,9,10].

The antibody conjugated NPs (ACNPs) approach is built on the success of nanotechnology and antibody therapies. When compared with the development of ADCs, ACNPs present many advantages, such as the delivery of the drug in a controlled manner, preservation of the chemical structure of the drug, reduced risk of secondary metabolites (if metabolism is unpredictable), and finally, potentially reduced toxicity [11]. Based on these features, the membrane proteins overexpressed in tumor cells can be used to design the antibodies that will later be implemented as the nanoparticle vector. The mechanism of action of both targeted therapies (ADCs and ACNPs) are very similar. After binding to the target, the complexes are internalized into the cell through receptor-mediated endocytosis [12], resulting in the formation of endosomes. Later, endosomes and lysosomes are coupled to release the drug into the cytoplasm [13]. However, the cargo of the nanoparticle can also diffuse directly through the cellular membrane, augmenting the cellular penetration of the compound.

Polymeric NPs can be generated from biodegradable and biocompatible raw materials, and the release of the drug can be controlled by the polymeric structure. Biodegradable and biocompatible polymers can be naturally or synthetically made. They are configured by ester, amide, and ether functional groups to easily break down in natural environments. Because polycaprolactone (PCL), polylactide (PLA), and poly(lactic-*co*-glycolic acid) (PLGA) are FDA-approved biocompatible polymers, they are the most used polymers for drug delivery systems (DDS) generation. Other biodegradable polymers such as poly(anhydrides) [14], poly(ortho esters) [15], poly(amides) [16], poly(phosphoesters) [17] and poly(alkyl cyanoacrylates) [18] are great candidates as raw materials to explore nanotechnology. However, their scarce commercial availability hampers their development.

The common ways to obtain biodegradable and biocompatible polyesters are polycondensation processes or physical aggregation. Both methodologies fail in controlling the molecular weight, polydispersities and stereoregularity of the polymer. The physicochemical properties of polymeric NPs as DDS, such as size, molecular weight, molecular weight distributions, crystallinity and polarity, will determine their efficacy and viability. Organometallic catalysis allows the preparation of polymers and copolymers with a precise control in their structure. Currently, structurally well-controlled polyesters, like PCL and PLA, FDA-approved polymers for DDS generation, are catalytically generated via ring opening polymerization (ROP) of lactone or lactide rings using organometallic catalysis. Nowadays, new promising strategies based on high technology and tailor-made building-block materials using chemical catalysis are used to obtain biodegradable and biocompatible polymers with controlled physical properties [19]. Recently, some examples of polyesters or polycarbonate architectures for DDS generation are being synthetized via ROP or ring opening copolymerization (ROCOP) [19,20]. These novel methodologies to generate tailor-made biocompatible polymers are more attractive than the traditional way, facilitating their implementation in the field of nanomedicine.

There are several methodologies to obtain polymeric NPs. All of them can be classified into two main strategies, top-down and bottom-up. In the top-down strategy the NPs are obtained from preformed polymers, while the polymerization is achieved during NP formulation in bottom-up strategies [21]. The most common methods for polymeric NP generation are top-down, multiple emulsion and nanoprecipitation. Such methodologies will interfere in the physical–chemistry properties of DDS, such as efficiency of encapsulation, loading-efficiency, morphology, biodegradation, average size, polydispersities, and surface charge. Whereas nanoprecipitation is more convenient for encapsulating hydrophobic drugs, multiple emulsion allows higher efficiencies for hydrophilic counterparts.

Concerning the release of the drugs from the polymeric NPs, in the first phase, the drug is released abruptly, followed by a second step ruled by diffusion through pores and channels of the NPs, ending with the degradation of the polymer. It is noteworthy to point out that the burst release is dismissed when NPs are prepared from stimuli-triggered polymers. Drug delivery systems (DDS) can be optimized for clinical applications by controlling these above-mentioned steps through modifications in the polymeric structure [22,23]. Modifications on the polymeric NPs can produce several advantages: (1) control of the cargo and the release of the drug, (2) prolongation of their stability in blood circulation, (3) an increase in their capacity to carry high toxicity drugs, and (4) overcoming drug resistance mechanisms.

Although a vast array of materials has been used to formulate ACNPs, this review focuses particularly on polymeric ACNPs for breast cancer therapy.

## 2. Polymeric Antibody Conjugated Nanoparticles (ACNPs) to Improve Treatments in Breast Cancer

Polymeric NPs appear to be the most promising drug carriers due to their nanoscale size and potential for selective targeting and controlled drug release [24,25,26,27]. NPs in the range of 100–400 nm have been widely reported to accumulate at the tumor site through the enhanced permeability and retention (EPR) effect [28,29]. This favors high accumulation of the drug, facilitating its delivery to the site of interest by convection and diffusion processes [30], reducing, at the same time, the damage to surrounding tissues [31,32]. The passive targeting is drastically influenced by the size and surface charge of the NPs. Both features are important for the retention of the NP in the tumor and for its circulation time.

After accumulation in the tumor region, drug efficiency can even be increased by the so-called active targeting. Binding to overexpressed receptors on target cells [33] that are poorly or not even expressed on normal cells [34,35,36,37] affords active targeting of NPs to tumor cells. Actively targeted NPs allow the delivery of the drugs at the desired location, avoiding normal tissues and, therefore, enhancing the therapeutic efficiency. For example, NPs bearing the monoclonal antibody trastuzumab, which interacts with the surface protein HER2, are expected to accumulate at sites where HER2 is overexpressed, such as in breast or gastric tumors. These particles can be loaded with other agents such as taxanes, which synergize with trastuzumab in the treatment of breast tumors. NP internalization occurs via receptor-mediated endocytosis [37,38]. Antibodies are the most frequently used ligands to target tumor cells [39,40]. Antibody fragments, such as antigen-binding fragments (Fab) may also be conjugated to reach higher diffusion rates and improve tumor uptake [41,42]. Trastuzumab is the most widely explored monoclonal antibody for generating novel nanomedicines for breast cancer therapy by far.

## 3. Conjugation Strategies for ACNP Generation

NP functionalization with antibodies or antibody fragments can be carried out mainly via the adsorption phenomenon, covalent-nature binding, or binding by the use of adapter molecules (Figure 1). The immobilization of antibodies into the nanoparticle surface must guarantee both the desired amount of antibodies per nanoparticle and the proper antibody orientation [43,44]. Additionally, the immobilization method must generate a stable bond and preserve the biological activity of the antibody [45].

### 3.1. Adsorption

This phenomenon is a non-covalent immobilization strategy that comprises physical adsorption and ionic binding [46] (see Figure 1A). Physical adsorption occurs via weak interactions such as hydrogen bonding, electrostatic, hydrophobic and Van der Waals attractive forces [47], while ionic binding occurs between the opposite charges of the antibodies and NP surfaces [48]. However, when compared to other methodologies such as covalent binding, adsorption provides less stability [49]. On the other hand, the fact that the interaction is non-covalent may allow release of the antibody in the tumor tissue, allowing deploy of its antitumoral properties. In this regard, three modified methodologies of functionalized PLGA NPs with trastuzumab were generated via adsorption, charged interactions and covalent binding. The results obtained via covalent binding demonstrated its higher stability versus charge interactions and adsorption [50].

### 3.2. Covalent Strategies

Covalent binding requires prior activation of the NPs [51]. Commonly, covalent strategies occur via carbodiimide chemistry, maleimide chemistry or “click chemistry”. Table 1 compiles a collection of works for breast cancer therapy based on covalent strategies for antibody bio-conjugation, including bio-conjugation using aptamers.

#### 3.2.1. Chemistry of Carbodiimide

The most common covalent conjugation corresponds with the binding through the primary amines of the antibodies [52] (see Figure 1C), without performing any chemical modification [53]. Previously, carboxyl groups of the NPs must be activated by the addition of cross-linking agents, where 1-ethyl-3-(-3-dimethylaminopropyl) carbodiimide (EDC) is the most used [54,55]. Even though it is not required, *N*-hydroxysuccinimide (NHS) or *N*-hydroxysulfosuccinimide (sulfo-NHS) are usually added to improve the coupling efficiency [56,57].

This methodology is easy but the coupling between functional groups and crosslinkers is not selective. This absence of control over antibody orientation into the NP surface is a primary disadvantage [58,59].

ACNPs prepared via this methodology have been used for detection and treatment of breast cancer. Table 1 shows a few works based on polymeric ACNPs where the antibody was conjugated via carbodiimide chemistry. The polymers mainly used were biodegradable polymers approved by the FDA, such as PLGA. Trastuzumab was the most frequently used antibody in these works [59,60,61,62,63,64,65,66]. As a representative example, monoclonal antibodies recognizing the specific profile of the cytokeratins expressed by breast cancer cells were conjugated to modify PLGA-based NPs via carbodiimide chemistry. Such ACNPs effectively delivered drugs into specific cells [67]. Acharya et al. prepared biodegradable PLGA NPs loaded with rapamycin and conjugated with cetuximab. These ACNPs were able to recognize the extracellular ligand-binding domain of epidermal growth factor receptor (EGFR). The latter receptor is expressed in a subset of breast tumors [68]. The therapeutic effects of docetaxel were greatly enhanced by the formulation of cetuximab conjugated ACNPs. The nanoparticle system demonstrated ~200 fold higher efficiency for the MDA MB 468 and MDA MB 231 cell lines respectively, which express EGFR [69,70].

#### 3.2.2. Maleimide Chemistry

This approach occurs through sulfhydryl groups (–SH) of antibodies (Figure 1C). Sulfhydryl groups are much less abundant than primary amines or thiols. They are present in the side chain of cysteine [71]. Free sulfhydryl, the de-protonated form of thiol at physiological pH, is a better nucleophile than thiol groups [72]. The –SH groups are generated on the antibodies via reaction with primary amines or by reduction of native disulfide bonds of antibodies [73,74]. Furthermore, this chemical group can be easily obtained by modification of the ε-amino of lysine residues with sulfhydryl-addition reagents. 2-Iminothiolane (Traut’s reagent) and *N*-succinimidyl S-acetylthioacetate (SATA) are the most common [75].

The reaction of maleimide groups towards free sulfhydryls is a thousand times faster than with primary amines at neutral pH, which improves selectivity [76,77]. A stable thioether linkage is formed after alkylation reaction with sulfhydryl groups [78,79] (see Figure 1C). The most used maleimide cross-linking reagents are the NHS/maleimide heterobifunctional linkers, PEGylated analogues (NHS-PEG-maleimide), succinimidyl 4-(*N*-maleimidomethyl)cyclohexane-1-carboxylate (SMCC) and sulfosuccinimidyl 4-(*N*-maleimidomethyl)cyclohexane-1-carboxylate (sulfo-SMCC) [80,81,82]. NHS-PEG-maleimide has been the most extensively used crosslinker in antibody conjugation strategies in polymeric NPs for breast cancer treatment (Table 1). SMCC is water-insoluble [83] but sulfo-SMCC is soluble in water due to a negatively charged sulfonate group on the NHS ring [84]. Therefore, using sulfo-SMCC as crosslinker, the free sulfhydryls of the antibody can be linked to the primary amines on the surface of the NPs via maleimide chemistry [85]. In this way, trastuzumab was attached to self-assembled chitosan-doxorubicin NPs via thiolation of lysine residues and subsequent linking of the thiols through sulfo-SMCC crosslinker to chitosan [86]. These ACNPS discriminated between HER2+ and HER2− cells in their mechanism of action. Heterobifunctional PEGylated linkers can also be used in the conjugation of thiolated antibodies to NPs through maleimide chemistry [87,88].

Non-selectivity of maleimide to cysteines has been reported. This lack of selectivity is due to exchange reactions with thiol-containing proteins in serum (e.g., albumin). Thus, non-homogenous conjugates and poorly defined yielding off-target cytotoxicity were obtained [89,90]. Antibody engineering techniques can be applied to increase homogeneity but the high cost and complexity of the strategy limits its use [3,91]. Alternative approaches to be explored are the use of next generation maleimides (NGM) or pyridazinediones and site-specific conjugation strategies (“click chemistry”) [92,93,94,95].

#### 3.2.3. Click Chemistry

“Click chemistry” reactions occur efficiently at room temperature, under mild and in aqueous solvents. These reactions, which require no or minimal purification, resulted in irreversible chemical bonds with the absence of cytotoxic byproducts [96,97,98,99].

[3+2] Azide–alkyne cycloaddition (AAC) reactions catalyzed by copper (I) (CuAAC), strain-promoted [3+2] azide–alkyne cycloaddition (SPAAC) reactions, typical [4+2] Diels–Alder (DA), and inverse electron demand hetero Diels–Alder (iEDDA) reactions [100,101,102,103,104] configure the “click chemistry” strategy. The advantage of using azide and alkyne groups relys on their low existence in biological systems and inertness towards the majority of functional groups and biomolecules [105]. Functionalization of NPs with azide or alkyne moieties goes through EDC/NHS coupling or maleimide–thiol conjugation. In this context, NHS ester or maleimide must be present at one side of the linker, while azide or alkyne groups must be located at the other [106].

Cu(I) catalysts accelerate AAC reactions [107,108]. However, the toxicity associated with these catalysts holds back their use in living systems [109]. Thus, CuAAC reactions were used to develop a HER2 targeted ACNP for delivering DOX to breast cancer cells [110]. These ACNPs included a pH sensitive block polymer, poly(L-histidine). A NHS-PEG-alkyne linker was introduced to trastuzumab to provide an alkyne group into the antibody and thus further improve cellular uptake and toxicity for MCF7 and SKBR3 breast cancer cells. In the same way, optimum polyethylenglycol (PEG) coverage over trastuzumab conjugated ACNPs was also prepared via CuAAC conjugation to improve the pharmacokinetics [111]. Of note, developments in copper-free variants of AAC were achieved by introducing a strain-promoted azide–alkyne cycloaddition (SPAAC) reaction to overcome the CuAAC limitations [112,113,114].

The SPAAC variant introduces ring-strained alkynes (cycloalkynes) to create stable triazoles [115]. Conjugated PLGA–PEG NPs with Fab fragments of trastuzumab or cetuximab for the treatment of breast cancer were obtained via SPAAC [89]. The evaluation of these ACNPs using HER2+ breast cancer lines (HCC1954) demonstrated that site-specific conjugation by SPAAC increased the antigen binding capacity and the conjugation efficiency [89].

Shi et al., developed amphiphilic furan-functionalized self-assembling copolymers (poly(TMCC-*co*-LA)-*g*-PEG-furan) to form polymeric NPs and DA cycloadditions were used to conjugate anti-HER2 antibodies [116]. On the other hand, polymeric ACNPs comprised of surface furan groups were used to bind, by DA coupling chemistry, both anti-HER2 antibodies and chemotherapeutic doxorubicin for intracellular delivery of doxorubicin [117]. The cytotoxicity, specificity and intracellular uptake were ascertained in SKBR cells with this type of ACNP.

Thereafter, Logie et al. designed ACNPs of docetaxel to improve the efficacy in a NOD-SCID- IL-2Rgnull mouse model of breast cancer [118]. These ACNPs were generated with the incorporation of a novel HER2 fragment antibody, Fab 73J. The lower toxicity of the ACNPs successfully allowed for higher dosing regimens.

### 3.3. Binding by Adapter Molecules

Non-covalent approaches by using adapter biomolecules avoid randomly oriented antibodies [119]. The most relevant binding strategy exploits the biotin–avidin interaction, which is based on the strong binding affinity between biotin and a biotin-binding protein, such as avidin or its analogues [120,121] (See Figure 1B). 

Biotin is a potential tumor-targeting moiety because it is a water-soluble vitamin (vitamin H), it is essential for growth and cell signaling, and their receptors are overexpressed on a broad range of cancer cells [122,123,124]. As an example of this non-covalent approach, the biotin-decorated PEG-PCL NPs were co-loaded with doxorubicin and quercetin to overcome doxorubicin-resistant breast cancer. In the same way, 7-ethyl-10-hydroxycamptothecin (SN-38) was successfully loaded into PLGA-PEG-biotin NPs to treat breast cancer. Compared to non-biotin conjugated NPs, these conjugated NPs increased the in vivo antitumor drug efficacy [125,126].

On the other hand, three *N*-acetyl glucosamine and four mannose residues constitute the oligosaccharide side chain of the tetrameric glycoprotein avidin [127,128]. Each of the four avidin monomers can bind to biotin [129]. However, because avidin is a basic protein with a high isoelectric point (pI) (~10.5) [130], both positively charged residues at physiological pH and glycosylation may lead to nonspecific binding of avidin to other molecules [131]. Thus, other forms of avidin, like streptavidin or neutravidin, are preferred [132]. Contrary to avidin, streptavidin is not a glycoprotein [133,134] and its pI is much lower, which prevents interactions with sugar receptors [135]. One of the most elegant strategies applied consisted in engineering monomeric avidin/streptavidin to monovalent biotin binding [136] and nanosized poly-avidins named Avidin-Nucleic-Acid-Nano-Assemblies (ANANAS) [137]. Notably, the ANANAS technology along with the use of an antibody (cetuximab) to target a ligand demonstrated an in vivo therapeutic efficacy at a dose of doxorubicin below the one clinically used [138].

To minimize the immunogenicity, a recombinant non-glycosylated form of avidin (neutravidin), was developed [139,140,141]. Warlick et al. prepared anti-HER2-modified NPs by the use of biotin/neutravidin aptamers to efficiently bind to and internalize in HER2-overexpressing cells. Trastuzumab was biotinylated and human serum albumin NPs were functionalized with neutravidin via NHS-PEG-Mal heterobifunctional crosslinker conjugation. This work clearly proved ACNPs combine specific tumor targeting with the drug delivery properties [141].

Finally, the use of aptamers was extended by Powell et al. into the development of a targeted delivery of siRNA specific to the multidrug resistance (MDR) gene of metastatic breast cancer. The use of a cholesterol, DOTAP and PLGA or PLGA-PEG lipid-polymer hybrid liposomes nanocarrier system overcame resistances of breast cancer cells generated against therapeutic drugs. The mechanism consists in enhancing the knockdown of the MDR gene (P-gp) by the use of aptamer-labeled P-gp siRNA encapsulated NPs [142].

## 4. Challenges for Clinical Implementation

The myriad of antibody conjugation strategies to generate polymeric ACNPs, along with the elicited benefits, in terms of site-selectivity and decreased off-target cytotoxicity, open attractive opportunities for the development of novel therapeutics in breast cancer.

There are several challenges in the clinical development of this kind of therapeutic agent. Firstly, identification of a suitable target and availability of a specific antibody play a significant role. The design of an effective antibody against a protein that is overexpressed in the tumor cell, but not in non-transformed tissue, is desirable. This finding has been shown to increase the therapeutic index of ADCs and should also apply to ACNPs [143]. However, not many proteins have been described as overexpressed in breast cancer. Table 2 shows a list of ADCs in clinical development. Of note, most of the antibodies used to build these ADCs were not designed to face overexpressed proteins due to gene amplification, as most of them are immunotherapy-based antibodies. In this regard, ADCs can be based on antibodies that lack tumor specificity, like the recently approved ADC, sacituzumab-govitecan, against the trophoblast cell surface antigen 2 (Trop-2) [144,145]. This protein is not only expressed in triple negative breast cancer but in many other epithelial tumors [145]. Nonetheless, this ADC is still active as it maintains an adequate therapeutic index. Indeed, the Trop-2 protein is mainly expressed during germline differentiation and this type of protein is generally expressed in adult mature tissue but not in tumors [145].

Considering the difficulty in finding specific proteins within the tumor, in the case of ADCs, the diffusion of the payload to target not only tumor cells but also the tumor stroma, affecting the tumor environment, emerges as a desirable effect. This is the so-called bystander effect and this has been demonstrated as key for the development of some therapeutics like trastuzumab-deruxtecan, an ADC with a cleavable linker against HER2 [146]. In addition, the bystander effect can revert resistance caused by cancer cells that do not express the target protein [147,148]. ACNPs can be designed to inherently possess a bystander effect, therefore reverting drug resistance due to tumor heterogeneity. In line with this, resistance to some ADCs has been described as related to the lysosomal processing of the antibody [149,150,151]. In this context, ACNPs could overcome this limitation. Finally, polymeric NPs own the ability to release the payload when a specific pH, mainly acidic, is produced in the tumor, particularly in areas of poor vascularization, therefore acting on areas where cancer drugs have limited penetration [152].

We envision the next steps will be developing new strategies to target specific proteins of non-tumor components of the tumor, including vasculature and the immune system. As it is shown in Table 2, a number of ADCs are formed by antibodies that recognize proteins expressed at the membrane of immune system cells, including anti-LAG3, PD-L1, CD73 or TGF ß, among others. A number of these membrane proteins are involved in the stimulation of an immunosuppressed environment. ACNPs aimed at targeting these immune system proteins, combined with the capability of these DDS to load antitumor drugs, opens the possibility of using a single ACNP to carry out multiple antitumor effects. The latter may substantially increase the antitumor capability and may therefore represent a novel and efficient way of fighting tumors. In some specific subtypes of breast cancer, the expression of a particular protein has shown to be a remarkable target. This is the case for HER3 in luminal and HER2 resistant tumors, where recently published data shows signs of potential activity.

Other strategies are being explored for the generation of more efficient ACNPs. Redox responsive polymeric ACNPs for the treatment of breast cancer have been designed [63]. These ACNPs encapsulated doxorubicin and were decorated by folic acid, trastuzumab and folic acid plus trastuzumab, all of them based on carbodiimide chemistry. By comparing in vivo studies, no significant differences in cytotoxicity were observed between ACNPs formulations, and no toxicity to heart, liver or kidney were reported when compared to free doxorubicin.

The influence on the ACNP shapes are still at an early stage and further investigations are required to determine the effects of NP shape on cellular uptake. As a representative example, worm-like PCL-PEG ACNPs for the controlled release of paclitaxel to HER2+ breast cancer cells were designed [153]. A new conjugation strategy based on the use of –C–N– to covalently bond trastuzumab to the surface of NPs was claimed to improve the stability of the formulation in physiological circulation. In this sense, it is worth noting the use of ultrasound to improve the cell targeting capability in vitro and efficiency in vivo of the paclitaxel-loaded ACNPs in a very recent work [66]. The combination of ACNPs and ultrasound enhanced the paclitaxel targeting and accumulation in breast cancers.

Finally, the use of two different antibodies as a full molecule or using only the Fab fragment working as a bispecific antibody could redirect the nanoparticle to two different targets. For instance, one antibody to a protein contained at the membrane of a tumor cell and the other to an immune suppressive target also expressed in a cancer cell. On top of this dual mechanisms, we will also have the effect of the payload. An example will be the use of an anti-HER2 in combination with an anti-PD-L1 in the same nanoparticle.

## 5. Conclusions and Outlook 

We envision two current challenges: First, the identification of the best polymeric structure to better modulate the payload release profile in a given biological context.

Second, the best choice of the antibody and conjugation strategy to better guide the nanoparticle to the tumor area. Optimization of these two features can improve the pharmacokinetic profile, facilitate target binding, induce a bystander effect, and produce direct antitumor activity in addition to immunologic activation. We are aware that countless information will arise in the future in this field providing evidence of the best development process.

Table 1 and Table 2 are listed below.

## Figures and Tables

**Figure 1 pharmaceutics-12-00802-f001:**
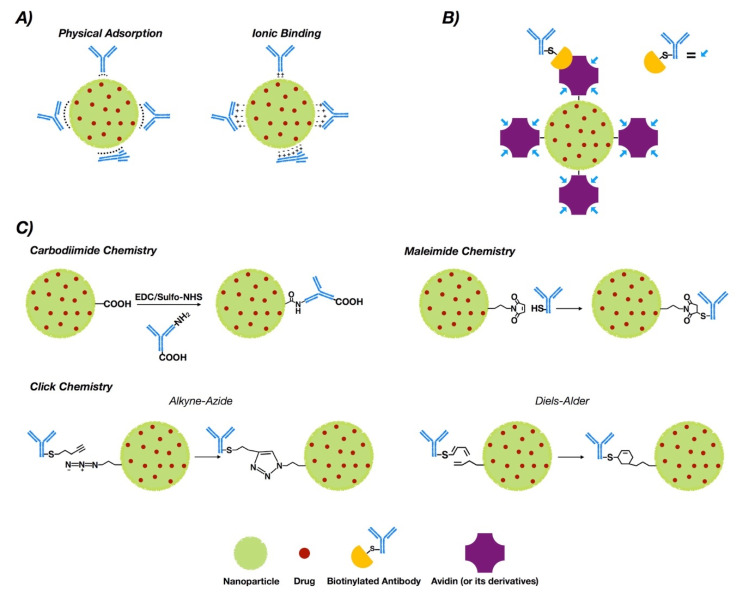
Conjugation strategies for antibody conjugated nanoparticle (ACNP) generation: (**A**) adsorption; (**B**) use of adapters; and (**C**) covalent binding.

**Table 1 pharmaceutics-12-00802-t001:** Main reported works on antibody conjugation strategies of polymeric NPs for the breast cancer treatment.

Conjugation Method	Nanoparticle Characteristic	Drug Encapsulated	Antibody Attached (Cross-Linker and or Substrategy)	Reference
Carbodiimide Chemistry	Magnetic poly(d,l-lactide-*co*-glycolide)	Doxorubicin	Trastuzumab	[59]
	Poly(lactic-*co*-glycolic acid) (PLGA)		Anti MCF-7 mAb	[67]
	Poly(d,l-lactide-*co*-glycolide) (PLGA)	Rapamycin	monoclonal EGFR antibody	[69]
	Poly(lactide)-d-α-tocopheryl polyethylene glycol succinate (PLA-TPGS/TPGS–COOH) copolymers	Docetaxel and iron oxides	Trastuzumab	[60]
	Poly(lactide)-d-α-tocopheryl polyethylene glycol succinate (PLA-TPGS/TPGS-COOH) copolymers	Docetaxel	Trastuzumab	[61]
	Amine terminated d-α-tocopheryl polyethylene glycol 1000 succinate (TPGS–NH_2_)	Docetaxel	Cetuximab	[69]
	Poly(N-vinylpyrrolidone/Poly(d,l-lactic-*co*-glycolic acid) (PVP–PLGA)	Tamoxifen	Trastuzumab	[62]
	Folate conjugated poly(d,l-lactide) (PLA) polyethylene glycol (PEG) PLA-PEG-PLA-urthane-s-s random multiblock copolymer	Doxorubicin	Trastuzumab	[63]
	(PEG-β-PLGA) copolymer	Indocyanine green and doxorubicin	Anti-HER2	[70]
	Poly(lactic-*co*-glycolic acid) (PLGA)	Epirubicin	Trastuzumab	[64]
	Poly(ethylene glycol)-poly(ε-caprolactone) copolymer (PEG-PCL)	Dasatinib	Trastuzumab	[65]
	Poly(lactic-*co*-glycolic acid) (PLGA)	Paclitaxel	Trastuzumab	[66]
Maleimide Chemistry	Human Serum Albumin (HSA)		Trastuzumab (NHS-PEG5000-Mal)	[154]
	Poly(2-methyl-2-carboxytrimethylene carbonate-*co*-d,l-lactide)-graft-poly(ethylene glycol)-furan (Poly(TMCC-*co*-LA)-*g*-PEG-furan)	Doxorubicin	Trastuzumab	[118]
	Poly(d,l-lactide-*co*-glycolide) (PLGA)	Docetaxel	Trastuzumab (NHS-PEG-Mal)	[155]
	Chitosan	Doxorubicin	Trastuzumab (sulfosuccinimidyl 4-(N-maleimidomethyl) cyclohexane-1-1-carboxylate)	[86]
	Bovine serum albumin (BSA)	5-fluorouracil	PR81 (NHS-PEG7500-Mal)	[156]
	Poly(d,l-lactide-*co*-glycolide) (PLGA)	Paclitaxel	anti-CD133 (NHS-PEG-Mal)	[157]
	Poly(d,l-lactide-*co*-glycolide) (PLGA)	Paclitaxel	Trastuzumab (DSPE-PEG2000-Mal)	[88]
	Poly(d,l-lactide-*co*-glycolide) (PLGA)	Paclitaxel	Clone 6, AM6 (NHS-PEG-Mal)	[158]
	Polyethyelenimine-polyethylenglycol copolymer (PEI-PEG)	SiRNA	anti-HER2 Nb (RR4) (NHS-PEG3500-Mal)	[159]
Click Chemistry	(Poly(TMCC-*co*-LA)-*g*-PEG-furan)		Trastuzumab (Diels-Alder)	[117]
	(Poly(TMCC-*co*-LA)-*g*-PEG-furan)	Doxorubicin	Trastuzumab (Diels-Alder)	[118]
	(PLGA-β-PEG-azide and PLGA-β-PHis-β-PEG-azide)	Doxorubicin	Trastuzumab (CuAAC)	[111]
	Poly(d,l-lactide-*co*-2-methyl-2-carboxytrimethylene carbonate) (P(LA-*co*-TMCC)	Docetaxel	Fab 73J (Diels-Alder)	[109]
	(PLGA–PEG-azide)		Trastuzumab (2,5-Dioxopyrrolidin-1-yl 1-((1R,8S,9S)-bicyclo[6.1.0]non-4-yn-9-yl)-3,14-dioxo-2,7,10-trioxa-4,13-diazaoctadecan-18-oate/SPAAC)	[89]
	Poly(d,l-lactide-coglycolide)-β-polyethylene glycol (PLGA-PEG)		Trastuzumab (CuAAC)	[112]
Aptamers	Human serum albumin (HSA)		Trastuzumab (Biotin/Neutravidin)	[142]
	Methoxy poly(ethylene glycol)-b-poly(ε-caprolactone) and methoxy poly(ethylene glycol)-b-poly(ε-caprolactone)	Doxorubicin and quercetin		[116]
	(PLGA-PEG-biotin copolymer)	SN-38		[117]
	Poly(d,l-lactide-coglycolide) (PLGA) and Poly(d,l-lactide-coglycolide) (PLGA)-β-polyethylene glycol (PEG)	Doxorubicin and p-gp SiRNA	Aptamer A6 (DSPE-PEG-Mal)	[143]
	Avidin-Nucleic-Acid-Nano-Assemblies (ANANAS)	Biotin-PEG-Atto488, biotin- Hz-doxo and biotin-PEG-Hz-doxo	Biotin-PEG-cetuximab, Cetuximab-Atto488 and cetuximab-Hz-doxorubicin	[139]

**Table 2 pharmaceutics-12-00802-t002:** Summary of ongoing clinical trials evaluating ADCs (antibody drug conjugates) in breast cancer describing the target, agent and the clinical stage ent.

Breast Cancer Subtype	Clinical Trial	Agent	Target	Combinatorial Agent	Phase
Luminal	NCT03874325	Durvalumab	PD-L1	Anastrozole	II
	NCT03409198	Chemo + ipilimumab + nivolumab	CTLA-4/PD-1	Ipilimumab, nivolumab, pegylated liposomal doxorubicin and cyclophosphamide	II
	NCT03691311	Denosumab	RANKL	-	I
	NCT03879174	Pembrolizumab	PD-1	Tamoxifen	II
	NCT03051659	Pembrolizumab	PD-1	Eribulin Mesylate	II
	NCT03393845	Pembrolizumab	PD-1	Fulvestrant	II
	NCT03492918	Pembrolizumab	PD-1	-	II
	NCT03608865	Durvalumab and Tremelimumab	PD-L1 and CTLA-4	-	II
	NCT04251169	Pembrolizumab	PD-1	Paclitaxel	II
	NCT01491737	Pertuzumab and Trastuzumab	HER2	Anastrozole	II
	NCT04088032	Durvalumab	PD-L1	Abemaciclib and Anastrozole	I
	NCT03241810	Seribantumab	HER3	Fulvestrant	II
	NCT03132467	Durvalumab and Tremelimumab	PD-L1 and CTLA-4	-	I
	NCT00405938	Bevacizumab	VEGF-A	Anastrozole	II
	NCT03659136	Xentuzumab	IGF	Everolimus	II
	NCT02990845	Pembrolizumab	PD-1	Exemestane/Leuprolide	I/II
	NCT00022672	Trastuzumab	HER2	anastrazole	III
	NCT03099174	Xentuzumab	IGF	Letrozole/anastrozole	I
	NCT02971748	Pembrolizumab	PD-1	-	II
	NCT03051672	Pembrolizumab	PD-1	Palliative radiotherapy	II
	NCT02997995	Durvalumab	PD-L1	-	II
	NCT04243616	Cemiplimab	PD-1	Cemiplimab + Paclitaxel + Doxorubicin + Cyclophosphamide	II
	NCT04243616	U3-1402	HER3	-	I/II
HER2	NCT03321981	MCI-A-128	HER2	Trastuzumab/Chemotherapy	II
	NCT03983395	GBR 1302	CD3/HER2	-	I/II
	NCT03052634	RC48	HER2	-	Ib/II
	NCT03523585	DS-8201a	HER2	-	III
	NCT03529110	DS-8201a	HER2	-	III
	NCT03734029	DS-8201a	HER2	-	III
	NCT03052634	RC48	HER2	-	Ib/II
	NCT03944499	FS-1502	HER2	-	I
	NCT03262935	SYD985	HER2	-	II
	NCT03255070	ARX788	HER2	-	I
	NCT03032107	Pembrolizumab	PD-1	T-DM1	Ib/II
	NCT03747120	Pembrolizuab	PD-1	Paclitaxel, Trastuzumab and Pertuzumab	II
	NCT02605915	Atezolizumab	PD-1	Pertuzumab, trastumuzab	Ib
	NCT03199885	Atezolizumab	PD-1	Paclitaxel, Trastuzumab and Pertuzumab	III
	NCT03125928	Atezolizumab	PD-1	Paclitaxel, Trastuzumab and Pertuzumab	IIa
	NCT03726879	Atezolizumab	PD-1	Doxorubicin, Cyclophosphamide, Paclitaxel, Trastuzumab and Pertuzumab	III
	NCT03417544	Atezolizumab	PD-1	Pertuzumab, trastumuzab	II
	NCT04034823	KN035	PD-1	Trastuzumab/Docetaxel	II
	NCT03112590	IFN-Y	IFN-Y	Paclitaxel, Trastuzumab and Pertuzumab	II
	NCT03135171	Tocilizumab	IL-6	Trastuzumab and Pertuzumab	I
	NCT03364348	Utomilumab	TNF	Trastuzumab emtansine or Trastuzumab	Ib
	NCT03414658	Utomilumab	TNF	Vinorelbine, Avelumab, Utomilumab	II
TNBC	NCT02655822	CPI-444	A2AR	Atezolizumab (PD-LI)	I/Ib
	NCT03454451	CPI-006	CD73	Pembrolizumab CD-I), cm-444 (A2AR)	I
	NCT03251313	JS001	PD-1	Gemcitabine and cisplatin	I
	NCT03012230	Pembrolizumab	PD-1	Ruxolitinib OAK)	I
	NCT02890069	Spartlizumab	PD-1	Everolimus (mTOR), panobinosat (HDAC), LCL161 (apoptosis), QBM076 (CXCR2)	I
	NCT03250832	TSR-033	LAG-3	Anti-PD-1 antibodies	I
	NCT02646748	Pembrolizumab	PD-1	Itacitinib OAK), INCB050465 (P13K)	I
	NCT02947165	NIS793	TGFß	Spartalizumab (PD-1)	I/Ib
	NCT03549000	NZV930	CD73	Spartalizumab (PD-1), NIR178 (A2AR)	I
	NCT02838823	JS001	PD-1	-	I
	NCT02622074	Pembrolizumab	PD-1	nab-Paclitaxel, doxorubicin,	I
	NCT03292172	Atezolizumab	PD-L1	cyclophosphamide, carboplatin R06870810 (BET)	I
	NCT02936102	FAZ053	PD-L1	Spartalizumab (PD-I)	I
	NCT03579472	M7824	PD-Ll/TGFß	Eribulin	I
	NCT0280744	MCSI 10	CSF-I	Spartalizumab (PD-1)	I
	NCT024602.24	LAG525	LAG-3	Spartalizumab (PD-1)	I/II
	NCT03241173	INCAGN01949	OX-40	Nivolumab (anti-PD-1) and/or ipilimumab (anti-CTLA4)	I/II
	NCT035912.76	Pembrolizumab	PD-1	Pegylated doxorubicin	I/II
	NCT02628132	Durvalumab	PD-L1	Paclitaxel	I/II
	NCT02657889	Pembrolizumab	PD-1	Niraparib (PARP)	I/II
	NCT03356860	Durvalumab	PD-L1	Paclitaxel, epirubicin, cyclophosphamide	I/II
	NCT02513472	Pembrolizumab	PD-1	Eribulin	I/II
	NCT02484404	Durvalumab	PD-L1	Olaparib (PARP)	I/II
	NCT02708680	Atezolizumab	PD-L1	Entinostat (HDAC)	I/II
	NCT02734004	Durvalumab	PD-L1	Diaparib (PARD	I/II
	NCT02614833	IMP321	LAG-3	Paclitaxel	II
	NCT03394287	SHR-1210	PD-1	Apatanib (VEGFR)	II
	NCT03414684	Nivolumab	PD. 1	Carboplatin	II
	NCT02648477	Pembrolizumab	PD-1	Doxorubicin	II
	NCT03004183	Pembrolizumab	PD-1	SBRT and ADV/HSV-tk	II
	NCT02536794	Durvalumab	PD-LI	Tremelimumab (CTLA-4)	II
	NCT02752685	Pembrolizumab	PD-1	nab-Paclitaxel	II
	NCT03095352	Pembrolizumab	PD-1	Carboplatin	II
	NCT03184558	Pembrolizumab	PD-1	Bemcentinib (AXL)	II
	NCT02971761	Pembrolizumab	PD-1	Enobosarrn (selecdve androgen receptor modulator)	II
	NCT02554812	Avelumab	PD-L1	Utomilumab (CD 137)	II
	NCT02849496	Atezolizumab	PD-L1	Olaparib (PARP)	II
	NCT03483012	Atezolizumab	PD-L1	Stereotactic radiosurgery	II
	NCT03164993	Atezolizumab	PD-L1	Pegytated doxorubicin	II
	NCT01898117	Atezolizumab	PD-L1	Carboplatin and cyclophosphamide or paclitaxel	II
	NCT02883062	Atezolizumab	PD-L1	Carboplatin	II
	NCT02981303	Pembrolizumab	PD-1	Imprime PGG (PAMP)	II
	NCT02819518	Pembrolizumab	PD-1	nab-Paclltaxel, gemcitabine, carboplatin	III
	NCT03498716	Atezolizumab	PD-L1	Paclitaxel, epirubicin. cyclophosphamide	III
	NCT03197935	Atezolizumab	PD-L1	nab-Paclitaxel, doxorubicin,	III
	NCT02425891	Atezolizumab	PD-L1	cyclophosphamide nab-Paclitaxel	III
	NCT03125902	Atezolizumab	PD-L1	Paclitaxel	III
	NCT02574455	Sacituzumab govitecan	Trop-2	-	III

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
