# Peer review of "An Overview of Antibody Conjugated Polymeric Nanoparticles for Breast Cancer Therapy"

_pharmaceutics, 2020, doi:10.3390/pharmaceutics12090802_

Round 1
Reviewer 1 Report
The manuscript entitled "An Overview of Antibody Conjugated Polymeric Nanoparticles for Breast Cancer Therapy" is an interesting overview of the use of antibody-conjugated nanoparticulate systems as drug nanocarriers for breast cancer treatment. The manuscript is adequately structured for its purpose and is easily readable and understandable, even for a non-expert reader. It contains an in-depth study about the state-of-the-art of the field, supported by an extense and current literature review. However, some points should be considered. Please, see the attached document for comments.

Author Response
We have carefully followed the referee comments and improvements have been made in the revised manuscript. We really want to thank the referee for the valuable suggestions. We are pretty sure they have improved the quality of our work.
Reviewer 2 Report
The review entitled ‘An Overview of Antibody Conjugated Polymeric Nanoparticles for Breast Cancer Therapy’ by Alberto Juan, Francisco Cimas, Ivan Bravo, Atanasio Pandiella, Alberto Ocaña, Carlos Alonso-Moreno is focused on antibody conjugated polymeric nanoparticles for breast cancer therapy. The authors shortly present the concept of polymeric nanoparticle preparation as well as active targeting, then reviewed conjugation strategies for immobilization of antibodies at the polymeric nanoparticle surface. Moreover, challenges for clinical implementation, conclusions, and outlook are presented. Generally, the topic is very interesting and in the scope of Journal, however, to improve manuscript I recommend to add the following points
*paragraph 1 (Introduction) please add information about polymeric nanoparticle preparation methods.
*References should be improved e.g.
The conjugation of nanoparticles with antibodies to generate so-called active targeted therapies was proposed early than citation suggests. For the general concept please cite an original paper where the concepts were proposed. Reference 2 is published in 2020, an active targeting concept was proposed earlier.
Author Response
A paragraph regarding common nanoparticle preparation methods have been included in the revised manuscript. Moreover, references have been updated following referee suggestion.
Reviewer 3 Report
- Authors described that “2. Polymeric ACNPs to improve treatments in breast cancer”, among the several strategies, ADC tech have more commercial product in the clinical. Readers would be like to understand what the difference between ACNPs and ADC.
- Authors may emphasize the passive and active target concept in the manuscript, it would be batter.
- The clinical trial information update carefully after this manuscript accepted
Author Response
A paragraph in the introduction section has been included to clarify the differences between ACNPs and ADCs. Moreover, passive and active target concepts have been explained in detail in the revised manuscript. Finally, clinical trials have been checked out before submitting the revision.
This manuscript is a resubmission of an earlier submission. The following is a list of the peer review reports and author responses from that submission.